# Multi-Frame, Lightweight & Efficient Vision-Language Models for Question Answering in Autonomous Driving

## Abstract

*Vision-Language Models (VLMs) and Multi-Modal Language Models (MMLMs) have become prominent in autonomous driving research, as these models can provide interpretable textual reasoning and responses for end-to-end autonomous driving safety tasks using traffic scene images and other data modalities. However, current approaches to these systems use expensive large language model (LLM) backbones and image encoders, making such systems unsuitable for real-time autonomous driving systems where tight memory constraints exist and fast inference time is necessary. To address these previous issues, we develop EM-VLM4AD, an efficient, lightweight, multi-frame vision language model which performs Visual Question Answering for autonomous driving. In comparison to previous approaches, EM-VLM4AD requires at least 10 times less memory and floating point operations, while also achieving higher BLEU-4, METEOR, CIDEr, and ROGUE scores than the existing baseline on the DriveLM dataset. EM-VLM4AD also exhibits the ability to extract relevant information from traffic views related to prompts and can answer questions for various autonomous driving subtasks. We release our code to train and evaluate our model here.*

## 1. Introduction

Vision-Language Models (VLMs) have emerged as powerful tools that possess holistic knowledge to solve tasks at the intersection of vision and language. This makes them a promising asset in autonomous driving (AD), allowing for a driver to interact with the VLM which can provide interpretable language representations of various driving safety tasks. Furthermore, VLMs can serve as end-to-end autonomous driving systems, eliminating integration and propagating errors between separate models specializing in specific sub-tasks of autonomous driving such as perception [14–16] and trajectory planning [25]. These potential benefits have propelled the development of many vision-language models and multimodal language models tailored for autonomous driving applications [5, 24, 31, 32, 38]. These models cover various aspects of autonomous driving including closed-loop control, perception tasks, and traffic agent behavior analysis.

Typically, the process in a VLM is the following: vision and text features are encoded separately, then fused together through a concatenation or projection layer, and then finally fed into an LLM to output some probability distribution over the vocabulary [37]. While generating text embeddings is relatively low-cost, the LM and image embeddings can often entail high computational costs. In real-time systems such as autonomous driving, prioritizing the development of VLMs with efficient inference times is crucial for practical deployment in vehicles. However, current research in applying multimodal language models to autonomous driving predominantly use large models such as BLIP-2 [20], GPT 3.5 [24], and LLaMA-7b [32], all of which contain over one billion parameters. Consequently, these models require expensive hardware and longer inference times, limiting their potential to be applied in current vehicles and accessibility for researchers with limited computational resources.

This paper focuses on the development of lightweight vision-language models with less than one billion parameters than can accurately and efficiently answer questions related to autonomous driving safety tasks. We develop the model EM-VLM4AD: Efficient, Multi-Frame Vision-Language Model for Autonomous Driving. We use the DriveLM dataset [31], which offers real, multi-view traffic scene images paired with question/answer pairs to train this model. Our contributions are as follows:

- We develop an efficient, smaller vision-language model EM-VLM4AD that consumes at least **10x** less memory and floating point operations (FLOPs) than current AD-VLMs, and can also respond to questions conditioned on multiple frames.
- We explore two different lightweight LM backbones for EM-VLM4AD: a finetuned Text-to-Text Transfer Transformer (T5) Base LM and an 8-bit quantized T5-Large LM finetuned using low-rank adaptation (LoRA) [18].

- We compare our model efficiency and performance on BLEU-4 (Bilingual Evaluation Understudy), CIDEr (Consensus-based Image Description Evaluation), ROUGE-L (Recall-Oriented Understudy for Gisting Evaluation), and METEOR (Metric for Evaluation of Translation with Explicit Ordering) to the baseline for the DriveLM dataset [31], demonstrating stronger performance in all metrics even with superior efficiency using a much smaller model.

## 2. Related Research

### 2.1. Vision-Language Models

Initially designed to operate on sequence data, Transformers [33] achieved state-of-the-art performance for natural language processing tasks. This propelled the development of Large Language Models, which learn general statistical properties of language through pretraining Encoder [9], Encoder-Decoder [29], and Decoder [2, 27, 32] Transformer architectures on a large corpus of tokens. These pretrained models can then be finetuned for downstream, more specialized language tasks. Dosovitskiy et al. [10] introduced the application of Transformers to image tasks with the Vision Transformer (ViT), which converts images into a sequence representation of image patches that can be processed by Transformers. Vision-Language Models bridge the gap between LLMs and Vision Transformers, encoding images and text into a combined latent representation and then utilizing cross-modal pre-training tasks to learn text and image correlations. This general approach to multimodal learning has sparked a variety of vision-language models. Radford et al. [28] devise a pre-training task of matching text captions with images to develop CLIP, which learns state-of-the-art image representations and exhibits strong zero-shot transfer capabilities for many image classification tasks. BLIP-2 [20] introduces a two stage pretraining process to train a Querying Transformer "QFormer" that serves as a intermediary between a frozen image encoder and language model. This approach outperforms much larger vision-language models such as Flamingo [1] and is capable of zero-shot image-to-text generation. Instruct-BLIP [7] builds off BLIP-2 and is a general-purpose VLM that aggregates public vision-language datasets and transforms them into an instruction tuning format. The VLM most similar to the model introduced in this paper is VL-T5 [6], which extends a pre-trained T5 to learn to generate text labels conditioned on a combination of a text and image embedding. Using a pre-trained LLM as a framework for multi-modal tasks harnesses the text generation ability of these models, critical for the question-answering task in our research. Despite their strong performance across many tasks, deploying these large models, which often exceed one billion parameters, is difficult for real-time applications [11]. Consequently, researching compression techniques like distillation [12, 21], quantization, and pruning is imperative to reduce VLM latency and computational costs.

### 2.2. Multimodal LLMs for Autonomous Driving

While autonomous driving systems mainly use visual features, introducing linguistic features can enhance the interpretability of these systems and even help identify novel traffic situations [13]. This benefit has sparked research interest in integrating multimodal data to train language models to become autonomous driving agents. Chen et al. [5] design an architecture that fuses vectorized numeric modalities with a pretrained LLaMA-7b [32] to solve Driving Question Answering tasks. Using a two-step training approach, they initially ground the vector representations into interpretable embeddings for the frozen LLaMA model, followed by finetuning the LLM with LoRA [18]. DriveGPT4 [38] also adopts LLaMA as a backbone LLM and CLIP as a visual encoder, using a traffic scene video and prompt text as input to generate answers and low-level vehicle control signals. To expand off the fixed and rigid QA labels from the BDD-X dataset [19], DriveGPT4 is trained on instruction tuning data generated by ChatGPT/GPT4. DriveGPT4 only uses a single-view camera, which restricts it to questions involving a single view. Wang et al. [35] introduce DriveMLM, which uses multi-view images, LiDAR Point Clouds, traffic rules, and user commands from a realistic simulator to perform closed-loop driving. This multimodal model is built from LLaMA-7B and ViT-g/14 as the image processor. Sha et al. [30] devise a chain-of-thought [36] framework for driving scenarios using ChatGPT 3.5 to provide interpretable, logical reasoning for autonomous driving systems. Mao et al. [24] also leverage the GPT-3.5 model to create a motion planner for autonomous vehicles. Their model, GPT-Driver, reformulates motion planning as a language modeling problem by representing planner inputs and outputs as language tokens. Recently, Sima et al. [31] released the DriveLM dataset, a Graph Visual Question Answering dataset that provides question-answer pairs related to perception, behavior, and ego-vehicle planning based off multi-view image data from the NuScenes dataset [4]. To introduce a baseline, Sima et al. finetune BLIP-2 [20] for this novel dataset.

While these approaches provide valuable explainability for AD systems and exhibit strong performance for end-to-end tasks, all these models use LLMs with over one billion parameters (GPT 3.5, LLaMA, etc.) and expensive image encoders like CLIP and ViT-g/14. This makes them primarily suitable for offline scenarios where latency is not a priority, but not for online situations where real-time inference is paramount.

## 2.3. Multi-Image Vision-Language Models

In the realm of autonomous driving, modalities beyond text and image such as LiDAR, radar, or video offer important features for many downstream tasks. However, most vision-language models are pre-trained for single-image single-text problems, making it unfeasible to directly input multiple images or modalities with a piece of text [37]. Consequently, it is necessary to consolidate multiple modalities and text into a single embedding that can be used by a VLM. DriveGPT4 [38] encodes video input by pooling CLIP visual encodings of each video frame. DriveMLM's [31] multimodal tokenizer uses QFormer to embed video and LiDAR data, and then concatenates these embeddings with text and system message embeddings. Wu et al. [37] find that using gated attention pooling across each individual image embedding helps introduce more non-linearity and extracts visual information across multiple images. Importantly, this gated attention method introduces a negligible amount of computational overhead, rendering it an ideal choice for our model to aggregate multi-view traffic scene images into a unified embedding.

## 3. Methods

Our model for Visual Question Answering (VQA) in Autonomous Driving, EM-VLM4AD, consists of a custom image embedding network and a pre-trained T5 language model [29]. We describe these following modules and the overall training process in this section.

### 3.1. Image Embedding Network

To tackle multi-view (Front, Front-Left, Front-Right, Back, Back-Left, Back-Right) QA tasks for autonomous driving, individual image embeddings need to be aggregated into a single embedding. This unified embedding can then be concatenated with a text embedding to serve as input to the LM. In typical VLMs, the image embedding process uses models like CLIP or object detection networks, resulting in a slow extraction process. To address this, we adopt the patch projection embedding scheme introduced by ViT [10]. Given an RGB image $I \in \mathbb{R}^{3 \times H \times W}$, the images are flattened and sliced into patches with a linear projection and positional embedding. This creates a latent image representation $V_i \in \mathbb{R}^{S_I \times H_I}$, where $S_I$ is the sequence length for the image embedding and $H_I$ is the hidden dimension of the image embedding. We use the pretrained weights of ViT-B/32 pretrained on ImageNet [8] to generate these image embeddings.

This leaves us with 6 distinct individual image embeddings from each view, which now need to be combined. We first flatten each image embedding into a one-dimensional vector and then use gated pooling attention as described by Wu et al. [37]. Given the individual image embeddings $V_i$,

gated pooling attention learns a single embedding:

$$V = \sum_{i=1}^{N} \alpha_i V_i \qquad (1)$$

in which $\alpha_i$ are weights for the ith image such that $\sum_{i=1}^{N} \alpha_i = 1$ that are calculated using:

$$\alpha_i = \frac{exp\{w^T(tanh(ZV_i^T) \otimes sigm(GV_i^T))\}}{\sum_{j=1}^{N} exp\{w^T(tanh(ZV_j^T) \otimes sigm(GV_j^T))\}} \qquad (2)$$

where $w \in \mathbb{R}^K, Z \in \mathbb{R}^{K \times M}, G \in \mathbb{R}^{K \times M}, M = S_I H_I$, and $K$ is a hyperparameter we set to 128. Gated pooling attention introduces non-linearity which helps pool visual information across the image. With this combined image embedding $V \in \mathbb{R}^{S_I \times H_I}$, we then project this embedding to match the embedding dimension $H_T$ of the text embedding so that we can concatenate the text and image embedding together with dimension $\mathbb{R}^{(S_T + S_I) \times H_I}$, where $S_T$ is the sequence length of the text embedding. This concatenated, multimodal embedding is then inputted into the LM to generate answer text.

### 3.2. Language Model

To mitigate the computational and inference costs of our vision-language model, we aim to use more lightweight LMs with less than one billion parameters. To achieve this, we use two different pre-trained versions of the T5 LM model: T5-Base, which contains around 223 million parameters, and an 8-bit quantized version of T5-Large ($\approx 750M$ parameters). Using these pre-trained LMs, we perform fine-tuning to adapt the LM to the concatenated multi-view image and text embeddings. In our experimentation, we found that fine-tuning the whole model for T5-Base works best, but for the quantized T5-Large we use LoRA-Fine-Tuning-Aware Quantization [22], which helps minimize quantization error with the initialization of LoRA weights.

### 3.3. Training Process

To train EM-VLM4AD, we use the DriveLM dataset [31], the most recent and comprehensive dataset for autonomous driving multi-view VQA with questions related to safety tasks such as perception, planning, prediction, and ego-vehicle behavior prediction. We use the training split of the DriveLM dataset, which contains 656 different scenes from NuScenes [4], 4,072 different multi-view frames, and 377,983 different multi-view/QA pairs. To evaluate our approach, we use a 90%/5%/5% split of the traffic scenes from DriveLM so we can evaluate how our model performs on unseen situations. Rather than train all components of our model in one stage, we use a two-stage approach as shown by Figure 1:

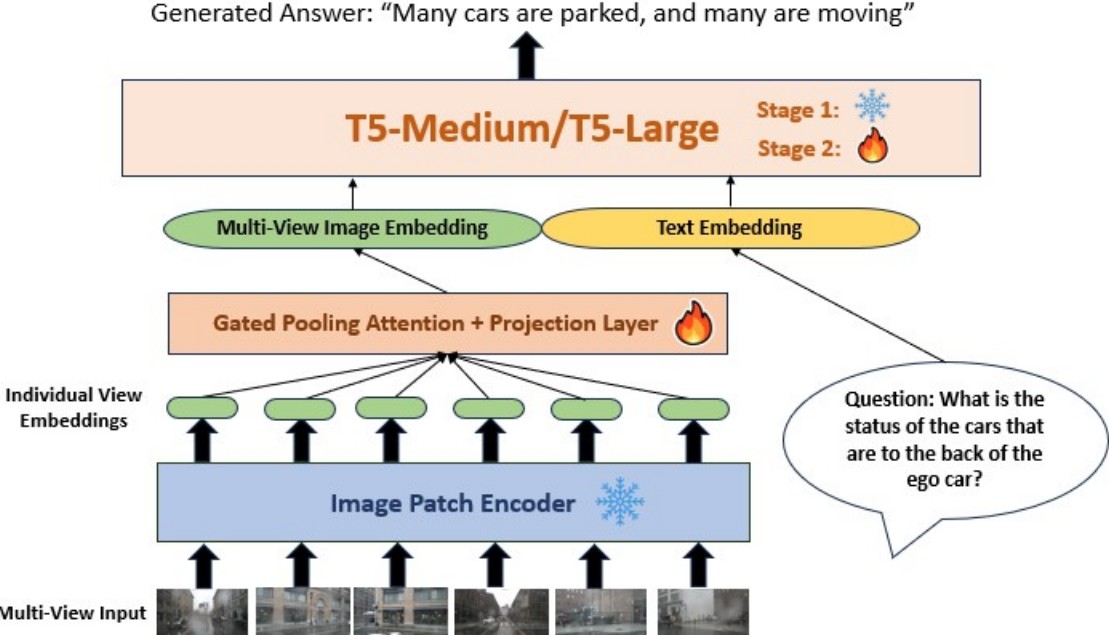

Figure 1. The diagram our model uses to respond to multi-view image input and question prompts. The T5 LM is frozen during Stage 1 of training so the image embedding network learns to align with the T5 embeddings. The image patch encoder is frozen throughout all stages of training, and the Gated Pooling Attention and Projection Layer is trained in both stages.

- In the first stage, we first freeze the image patch encoder and the LM and only train the gated pooling attention and projection layer. This forces the multi-view image embeddings to align with the type of embeddings the LM expects.
- Then in the last stage, we only keep the image patch encoder frozen and start to finetune the LM.

In summary, the image patch encoder is always frozen to maintain generalized image information gathered from pretraining, the gated pooling attention and projection layer is always trained, and the Language Model is only finetuned during the last stage of training.

We perform each training stage for six epochs, which takes around 2.5 days to finish for each model. We use a NVIDIA RTX 3090 Ti to train the T5-Large version of EM-VLM4AD and a V100 Google Colab instance to train EM-VLM4AD with T5-Base. We note that our models can be fit into a single T4 GPU instance, which allows to evaluate these models for free with Google Colab. For hyperparameters, we use a learning rate of 1e-4, weight decay of 0.05, an exponential learning rate scheduler, and a batch size of 4 for both approaches.

## 4. Experiments

This section presents an analysis of the quantitative, qualitative, and computational performance of EM-VLM4AD. We use the following metrics commonly used in image cap-

tioning tasks to assess the quality of the model-generated answers:

- BLEU-4 [26]: Measures how many 4-grams in the generated text match those in the reference text.
- ROUGE-L [23]: Calculates sentence similarity scores using the longest common sub-sequence between the generated text and ground-truth text.
- METEOR [3]: Considers exact matches, stemming, synonymy, and word order to measure alignment between model outputs and references.
- CIDEr [34]: To account for lexical and semantic similarity between the generated and reference text, CIDEr weights n-grams with their corresponding TF-IDF weight. This helps de-emphasize n-grams that commonly occur across all examples that may not have important meaning.

For computational analysis, we aim to analyze the memory and computational efficiency of our model, essential aspects in real-time systems where resource constraints exist and inference efficiency is paramount.

### 4.1. Quantitative Results

We evaluate the BLEU-4, ROUGE-L, METEOR, and CIDEr scores using the test set of unseen traffic scenes we create. Currently, the only existing approach on the DriveLM dataset is DriveLM-Agent [31], which is a finetuned version of BLIP-2. Since this model is not yet public and

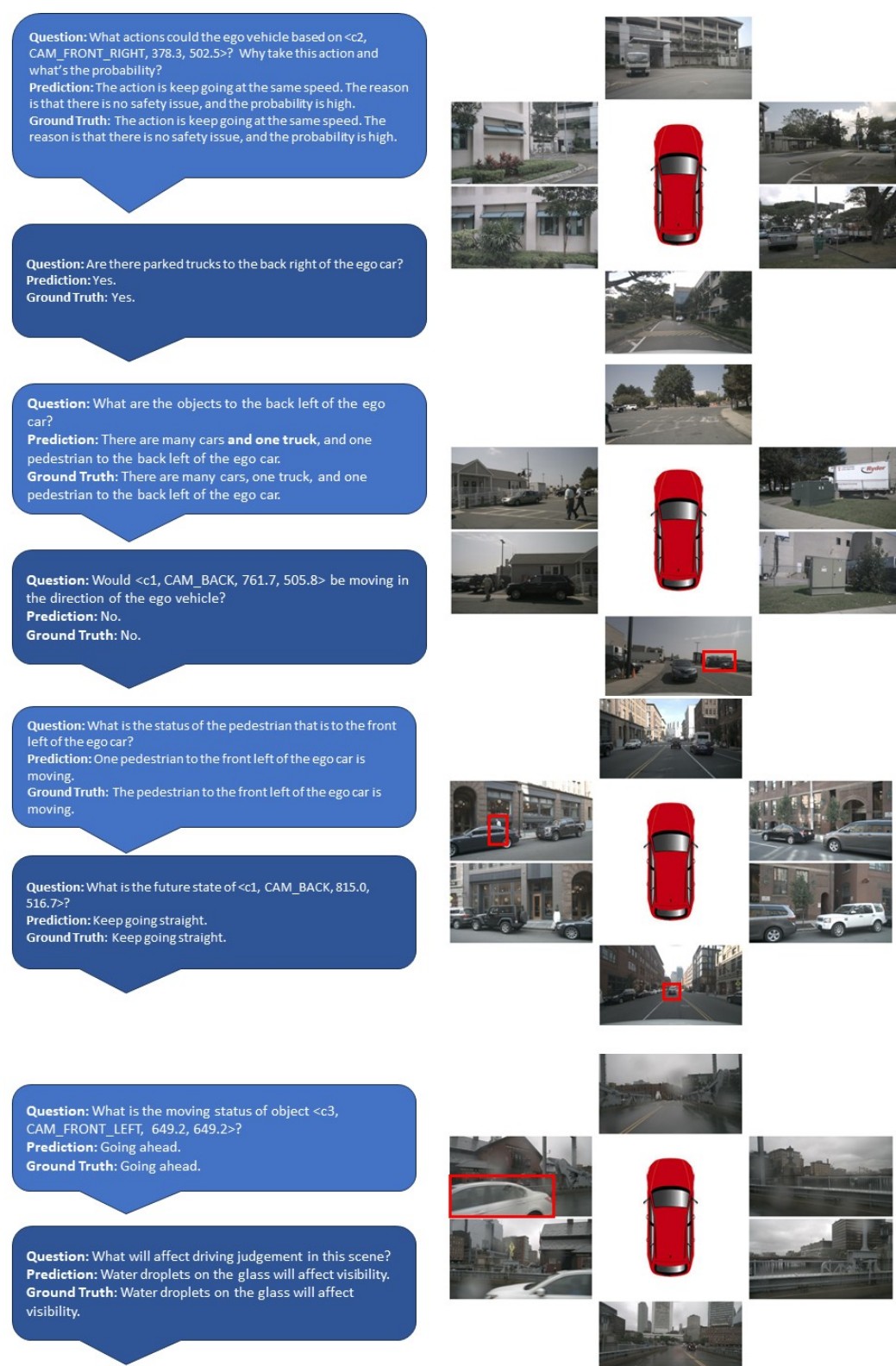

Figure 2. Example correct answer generations from EM-VLM4AD. As shown these in these examples, our model is able to perform VQA for various autonomous driving tasks such as perception, planning, and traffic agent behavior prediction.

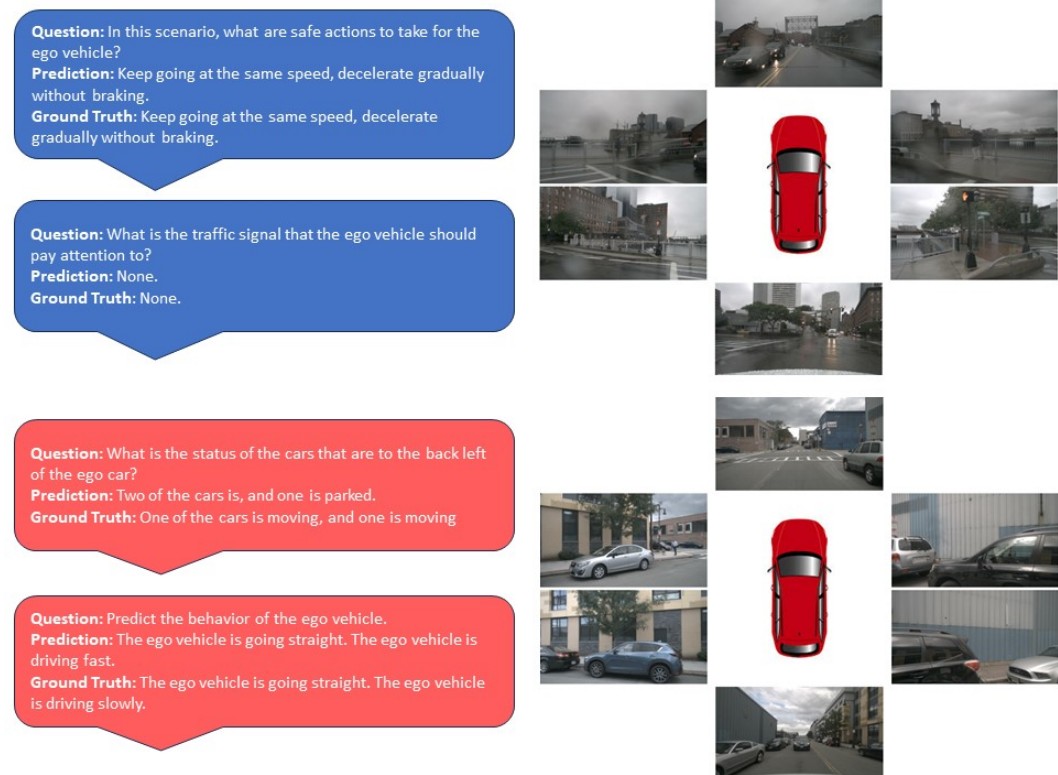

Figure 3. More example generations from EM-VLM4AD. As shown by the red QA examples, EM-VLM4AD can sometimes struggle with grammatical semantics and questions related to ego-vehicle behavior prediction, which may require video input for improved performance.

| Model | BLEU-4 ↑ | METEOR ↑ | ROUGE-L ↑ | CIDEr ↑ |
|---|---|---|---|---|
| EM-VLM4AD$_{Base}$ | **68.73** | **48.11** | **81.43** | **3.96** |
| EM-VLM4AD$_{Q-Large}$ | 67.86 | 47.64 | 81.00 | 3.90 |
| DriveLM-Agent [31] | 53.09 | 36.19 | 66.79 | 2.79 |

Table 1. Qualitative comparison of generated answers between DriveLM-Agent and EM-VLM4AD on their respective test sets. EM-VLM4AD$_{Base}$ uses a T5-Base LM backbone, while EM-VLM4AD$_{Q-Large}$ uses an 8-bit quantized T5-Large backbone. Both models outperform DriveLM-Agent in all statistics.

we do not have the computational resources to perform full-precision LoRA training of BLIP-2 , we benchmark our approach using the results Sima et al. [31] provide on their private evaluation set. The results from Table 1 demonstrate how both versions of EM-VLM4AD outperform DriveLM-Agent on all metrics, despite having at least 3 billion less model parameters. Out of all three models, the version of EM-VLM4AD that uses T5-Base is the top-performing model.

The superior performance of EM-VLM4AD with the T5-Base backbone over the 8-bit quantized T5-Large version can be attributed to the former's ability to train a larger parameter set. This facilitates a better adaptation of the language model to the input vision-language embeddings. Conversely, the LoRA finetuning approach for the

8-bit quantized T5-Large LM only changes 3.4% of the network's weights. While we did try full finetuning for the quantized LM, this over fine-tuned the LM and caused mode collapse.

The integration of multiple frames is a critical advantage that contributes to EM-VLM4AD's performance versus DriveLM-Agent. Unlike DriveLM-Agent, which only uses the front-view frame as input, our model successfully aggregates information across multiple views with our custom multi-view embedding network. Furthermore, while certain tasks done by LMs are defined as *emergent*, requiring larger models for sufficient results, our study underscores that learning to perform VQA on the DriveLM dataset can be done without increasing model complexity. Therefore, simply adding model complexity may not result

| Model | Pretrained Models Used | # of Parameters ↓ | FLOP Count ↓ | Memory (GB) ↓ |
|-------|------------------------|-------------------|--------------|---------------|
| EM-VLM4AD$_{Base}$ | T5-Base, ViT-b/32 patch embedder | **235M** | **9.47B** | 0.94 |
| EM-VLM4AD$_{Q-Large}$ | T5-Large, ViT-b/32 patch embedder | 769M | 31.5B | **0.77** |
| DriveLM-Agent [31] | BLIP-2 | 3.96B | 439B | 14.43 |
| DriveMLM [35] | LLaMA-7B, Vit-g/14 | 8.37B | 535B | 36 |
| LLM-Driver [5] | LLaMA-7B | 7B | 268B | 28 |
| Drive-GPT4 [38] | LLaMA 2, CLIP | 7.3B | 329B | 29.2 |

Table 2. Computational comparison of other LMs for Autonomous Driving with both versions of EM-VLM4AD. The EM-VLM4AD models have the smallest number of parameters, memory space, and FLOP count, making them the most efficient and computationally efficient VLM for autonomous driving.

in optimal improvements for this specific task.

## 4.2. Computational Analysis

We also perform computational analysis to see how EM-VLM4AD compares to other multimodal LMs for autonomous driving. Specifically, we focus on three key computational metrics: the # of parameters, # of Floating Point Operations (FLOPs), and memory in gigabytes (GB). For these methods, the image encoder and LM constitute the most computationally expensive aspects of these models, so we only focus on these two aspects when calculating these metrics. To estimate the FLOP count for each of these models, we use the fvcore FLOP counter module on examples from the DriveLM dataset with a A100 GPU. For the methods we compare to, we add the FLOPs of the image encoder and LM together. The results in Table 2 underscore that EM-VLM4AD is considerably more efficient than other methods, requiring less memory, computations, and model parameters. Notably, EM-VLM4AD with the T5-Base backbone has the least parameters and FLOP count, while EM-VLM4AD with the T5-Large backbone has the least memory requirements since model weights are only stored in 8 bits. These optimized model design choices enable EM-VLM4AD to provide fast inference times and require less computational resources, critical attributes for any LM implemented for real-time scenarios.

## 4.3. Qualitative Results

Figures 2 and 3 showcase some selected multi-frame answer generations produced by EM-VLM4AD. Our model can accurately respond to a variety of questions related to perception, traffic agent behavior identification, planning safe ego-vehicle behavior, and identifying important traffic elements in a scene. Through leveraging the general knowledge from the pretrained patch embedding network and T5-LM, our system can answer a wide spectrum of questions that encapsulate an end-to-end autonomous driving system. Additionally, EM-VLM4AD demonstrates the ability to understand the c-tag format employed by DriveLM, which encodes traffic objects as $< c, CAM, x_{pos}, y_{pos} >$. Moreover, this model learns to intelligently extract the most relevant frames for each question, making it an effective multi-frame VLM system. However, EM-VLM4AD exhibits two specific weaknesses: grammatical issues and issues answering questions related to behavior. EM-VLM4AD can occasionally generate answers with grammatical errors, hindering someone to understand the answer to a question. Adding training techniques such as distillation [17] with larger vision-language models, which have a better understanding of grammar rules, will help this smaller model learn these complex rules. EM-VLM4AD also struggles with behavior related questions, where the prompt is "Predict the behavior for the ego vehicle". Adding temporal context through inputting multi-view videos to our network would improve results on this type of question, since behavior related questions often need more than one frame to make accurate predictions.

## 5. Conclusion

We introduce EM-VLM4AD, a lightweight multi-frame vision-language model designed for Visual Question Answering across various autonomous driving tasks. Compared to other LMs tailored for autonomous driving, EM-VLM4AD exhibits notable advantages in terms of memory efficiency and computational requirements, and outperforms the reported scores of DriveLM-Agent in BLEU-4, METEOR, ROUGE, and CIDEr metrics on a DriveLM test dataset. EM-VLM4AD demonstrates proficiency in responding to a variety of autonomous driving questions and dynamically focuses on relevant camera views through our gated pooling attention layer, which effectively integrates view embeddings. In future research, we aspire to evolve our model into a video-language model capable of generating responses from multi-view video inputs, thereby enhancing EM-VLM4AD's ability to handle temporal-related inquiries. Furthermore, incorporating multimodal retrieval augmented generation to provide context can enable our model to extract insights from analogous traffic scenarios.

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
