# OpenReview forum: "Multi-Frame, Lightweight & Efficient Vision-Language Models for Question Answering in Autonomous Driving"
_thecvf.com/CVPR/2024/Workshop/VLADR — VLADR 2024 Poster_

### Official Review · Reviewer_LGmZ · 2024-04-21

**Rating:** 7
**Confidence:** 4

**Review:**

This paper introduces a few improvements to reduce the memory consumption and training/inference speed of Driving VisionLangauge Models (VLMs). It does so by: 1) removing the ViT encoder while keeping the patch embedding layer, 2) fusing embeddings of multi-cameras into a single embedding using a gated pooling attention method, 3) and using smaller LMs with optional quantization and LORA techniques.

This paper is highly relevant to our workshop and provides empirical results to support its claims. Thus I recommend acceptance.

---

### Decision · Program_Chairs · 2024-04-22

Accept (Poster)